# Polypharmacy and Its Association with Dysphagia and Malnutrition among Stroke Patients with Sarcopenia

**DOI:** 10.3390/nu14204251

**Published:** 2022-10-12

**Authors:** Ayaka Matsumoto, Yoshihiro Yoshimura, Fumihiko Nagano, Takahiro Bise, Yoshifumi Kido, Sayuri Shimazu, Ai Shiraishi

**Affiliations:** 1Department of Pharmacy, Kumamoto Rehabilitation Hospital, Kumamoto 869-1106, Japan; 2Center for Sarcopenia and Malnutrition Research, Kumamoto Rehabilitation Hospital, Kumamoto 869-1106, Japan; 3Department of Rehabilitation, Kumamoto Rehabilitation Hospital, Kumamoto 869-1106, Japan; 4Department of Nutritional Management, Kumamoto Rehabilitation Hospital, Kumamoto 869-1106, Japan; 5Department of Dental Office, Kumamoto Rehabilitation Hospital, Kumamoto 869-1106, Japan

**Keywords:** polypharmacy, sarcopenia, stroke, dysphagia, malnutrition, food intake

## Abstract

Evidence on polypharmacy in patients with sarcopenia is lacking. We aimed to examine the association of polypharmacy with swallowing function and nutritional risk in post-stroke patient with sarcopenia. This retrospective cohort study included hospitalized patients diagnosed with sarcopenia who needed convalescent rehabilitation following stroke onset. Study outcomes were the Food Intake Level Scale (FILS) as dysphagia assessment and geriatric nutritional risk index (GNRI) as nutritional status at hospital discharge, respectively. To examine the impact of admission polypharmacy, multivariate linear regression analyses were used to determine whether the number of drugs prescribed at hospital admission was associated with these outcomes. As a result, of the 586 patients enrolled, 257 (mean age 79.3 years, 44.0% male) were diagnosed with sarcopenia and were finally analyzed high admission drug prescription numbers were independently associated with FILS (β = −0.133, *p* = 0.017) and GNRI (β = −0.145, *p* = 0.003) at hospital discharge, respectively. Polypharmacy is associated with dysphagia and malnutrition in post-stroke patients with sarcopenia. In addition to the combination of nutritional and exercise therapies, review and optimization of prescription medications may be warranted to treat sarcopenia in order to maximize improvement in outcomes for these patients.

## 1. Introduction

Sarcopenia has become a growing concern in recent years and is a factor contributing to functional decline. Sarcopenia may lead to poor prognosis in older adults, including falls, fractures, decline in physical independence, dysphagia, and death. Furthermore, around 50% of patients who need convalescent rehabilitation present with sarcopenia [1], which is adversely associated with clinically important outcomes, such as improvement in activities of daily living (ADL), dysphagia, malnutrition, and decreased likelihood of discharge to home [2]. Sarcopenia is also negatively associated with improvement in physical function in patients after stroke [3,4]. Stroke often leads to decline in ADL [5], and many patients require rehabilitation [6]. Therefore, patients undergoing post-stroke rehabilitation would need an early diagnosis of sarcopenia and appropriate therapeutic intervention.

Dysphagia is an important problem in stroke patients. Dysphagia can be caused by diseases, such as stroke, but also by aging, muscle weakness, such as in the case of sarcopenia [7], cognitive decline [8], and drug side effects [9,10], and can lead to aspiration pneumonia and malnutrition. Dysphagia occurs in 27–64% of stroke patients, and 15% have residual dysphagia up to 1 month after stroke [11]. Dysphagia after stroke is associated with poor outcomes, including decreased ADL, low likelihood of discharge to home, and increased mortality [10]. In addition to assessing swallowing function and providing training to improve difficulty swallowing, chair-stand exercises [12], individualized nutritional support [13], and improved oral health [14] are reportedly associated with improved swallowing function in post-stroke patients. Hence, it is crucial to address factors associated with dysphagia and to intervene appropriately to improve physical function, ADL, and quality of life in post-stroke patients.

Polypharmacy is associated with poor patient outcome. The increased number of medications taken due to multimorbidity may increase the risk of adverse drug events, as well as drug–drug interactions [15]. Polypharmacy is associated with decreased physical and cognitive function [16], incidence of falls [17], and frailty [18] and can contribute to decreased ADLs. Many rehabilitation patients are subject to polypharmacy, especially stroke patients, who are at particularly high risk due to underlying diseases, and management of complications after stroke onset [19,20]. Polypharmacy is negatively associated with rehabilitation outcomes, such as ADL, cognitive function, and discharge to home [20,21,22]. Hence, it is important to address polypharmacy in patients undergoing rehabilitation by appropriately managing their medications. Furthermore, polypharmacy is associated with dry mouth [23] and affects oral and swallowing function. Among medications, antipsychotics [24] and anticholinergic burden are associated with the development of pneumonia [25] and dysphagia [26,27].

However, at present, little evidence is available on the effects of medications on dysphagia in rehabilitation patients, especially in patients with sarcopenia. Post-stroke patients with sarcopenia would have impaired swallowing function due to a decrease in muscle mass and strength, in addition to the effects of stroke. The current guidelines for the treatment of sarcopenia focus on exercise and nutritional therapy [28]; however, dysphagia interferes with the latter. Therefore, elucidating the association between medication numbers and dysphagia in patients with sarcopenia after stroke onset is clinically relevant to improve outcomes and to clarify the need to address polypharmacy in patients with sarcopenia.

Therefore, our study examined the effects of polypharmacy on dysphagia improvement in patients with sarcopenia who were undergoing convalescent rehabilitation after stroke.

## 2. Methods

### 2.1. Participants and Setting

We conducted a single-center study with retrospective cohort study design with patients who were admitted to a rehabilitation hospital with a total of 135 convalescent rehabilitation beds in Japan. All new stroke patients consecutively admitted to the convalescent rehabilitation beds were included in this study. The following patients were excluded from the study: refusal to participate in the study, altered consciousness (as indicated by the three digits of the Japan Coma Scale score) [29], implanted pacemakers that could interfere with measurement of bioelectrical impedance analysis (BIA), missing data, and transfer to other hospitals due to changes in condition during rehabilitation. Furthermore, among these patients, patients diagnosed with sarcopenia upon admission were exclusively included in the final analyses. Research period was from January 2015 to December 2021.

### 2.2. Data Collection

We recorded basic patient information, such as age, sex, body mass index, stroke type, and onset-to-hospitalization days. Swallowing function was assessed by speech therapists by the Food Intake Level Scale (FILS) [30]. Physicians assessed the severity of comorbidities by the Charlson comorbidity index (CCI) [31] and ADL before stroke onset with the Modified Rankin Scale (mRS) [32]. Dental hygienists assessed the oral function by the Revised Oral Assessment Guide (ROAG) [33]. Routine blood samples were taken immediately after admission to the hospital to measure blood levels of albumin, C-reactive protein, and hemoglobin. The risk of malnutrition was estimated using the Geriatric Nutritional Risk Index (GNRI) [22].

Physical therapists measured functional independence measure (FIM) within 3 days of admission and evaluated the patients for physical independence (FIM-motor), cognitive level (FIM-cognitive), and the sum of physical independence and cognitive level (FIM-total) [34].

During the first week of admission, nurses or registered dietitians estimated energy and protein intake by assessing visually the ratio of patients’ actual food intake to the amount of food served to patients [35]. Then, the nutritional intake per body weight was calculated by dividing each intake by the actual body weight of the patient upon admission.

Rehabilitation units per day were calculated by investigating the total units of physical therapy, occupational therapy, and speech therapy (1 unit = 20 min) received during hospitalization through a medical chart review and dividing by the length (days) of hospital stay.

### 2.3. Sarcopenia Definition

Sarcopenia was diagnosed at admission, by using hand grip strength (HG) and skeletal muscle mass index (SMI), based the Asian Working Group for Sarcopenia 2019 (AWGS2019) criteria [36]. Physical therapists measured the skeletal muscle mass using BIA and HG within 72 h of admission. HG of the non-dominant hand (or non-paralyzed hand in the case of hemiplegia) was measured using the Smedley hand dynamometer (TTM, Tokyo, Japan), and the largest of the three consecutive measurements was recorded. Body composition was measured using the InBody S10 (InBody, Tokyo, Japan) with standard protocols; the BIA is a validated instrument, where muscle mass estimates have been reported to be minimally affected by fluid overload [37]. The cutoff values specific for older Asian adults in the AWGS2019 were <5.7 and <7 kg/m^2^ for SMI and <18 and <28 kg for HG in women and men, respectively, and patients were diagnosed with sarcopenia when both SMI and HG were below the cutoff values.

### 2.4. Drug Information

Pharmacists investigated the number of medications through a review of medical charts. Upon admission, the ward pharmacists reviewed and listed the patients’ drug information. Temporary prescription medications, such as cold remedies, antibacterial agents for infections, medications used as needed for temporary symptoms, and over-the-counter medications, were excluded from the analysis, and only medications taken on a daily basis were counted. Polypharmacy was defined as prescribing five or more medications [15], and potentially inappropriate medications (PIMs) were counted based on the American Geriatrics Society’s 2019 Beers criteria [38], which are potentially inappropriate in older adults.

### 2.5. Rehabilitation Program

Convalescent rehabilitation programs tailored to each patient’s physical functions, disabilities, and ADL were implemented [39]. Rehabilitation was provided for a maximum of 3 h per day based on the national medical insurance. Physical therapy included, for example, range of motion training of joints in the extremities, facilitation of paralyzed limbs, basic movement training, walking training, resistance training, ADL training, etc. [40,41].

Medication management was performed by pharmacists and other multidisciplinary staff. Polypharmacy and inappropriate medications were screened, medication adherence was assessed, and adverse drug effects were observed. Medications that may affect rehabilitation, level of consciousness, nutritional status, or physical function were appropriately managed by reducing, discontinuing, or switching to alternative medications [42].

Nutritional management was provided under the supervision of dietitians and multidisciplinary team, such as nutrition support team, depending on the nutritional status and rehabilitation practices of each patient. For example, malnourished patients were provided with a high-energy, high-protein diet, and supplements, while obese patients were provided with caloric restriction and adequate protein to maintain muscle mass [13,43].

Oral management for each patient during the hospital stay was also tailored to the needs of each patient. Inpatient oral management included screening, assessment, and counseling of oral conditions, dental care, patient and family education, oral and dysphagia rehabilitation, oral care, and dietary assistance [44,45].

### 2.6. Outcomes

Primary outcome was FILS (dysphagia status) at discharge. FILS is a 10-point scale for the level of feeding and swallowing status where a score of 1–3: no oral intake, 4–6: combined oral intake and alternative nutrition, 7–9: oral intake only, and 10: normal. The secondary outcome was the GNRI (nutrition status) at discharge. GNRI is a nutrition screening tool that is calculated as (1.489 × albumin [g/dL]) + (41.7 × [weight/ideal weight]), where GNRI < 82 is severe nutritional risk, 82 ≤ GNRI < 92 is moderate nutritional risk, 92 ≤ GNRI < 98 is mild nutritional risk, and 98 ≤ GNRI is no risk.

### 2.7. Sample Size Calculation

Using the Power and Sample Size software, the sample size was calculated based on data from a previous study [46]. The results showed that the FILS scores of hospitalized patients after stroke were normally distributed with a standard deviation (SD) of 2.97. A true difference of 2 between the means of patients with and without polypharmacy would require a sample size of at least 49 patients in each group with a power of 0.9 and an alpha error of 0.05 to reject the null hypothesis, supporting the validity of our results.

### 2.8. Statistical Analysis

Results were reported as mean (SD) for parametric data, median and 25–75th percentiles (interquartile range [IQR]) for nonparametric data, and numeric (%) for categorical data. *p* < 0.05 was considered statistically significant. SPSS version 21 (IBM, Armonk, NY, USA) was used for all analyses. Bivariate analysis was performed with and without polypharmacy, and patients were divided into two groups. Comparisons between the two groups were made, depending on the type of variable data, using *t*-tests (two independent variables that were normally distributed), Mann–Whitney U tests (two independent variables that were not normally distributed), and chi-square tests (nominal variables).

Multivariate linear regression analyses were used to analyze whether the number of prescribed medications at admission was independently associated with FILS and GNRI at discharge, respectively. Confounders selected to adjust multivariate analysis for bias for each outcome were age, gender (male), stroke type, FIM-motor, FIM-cognitive, energy intake, ROAG, HG, SMI, CCI, rehabilitation therapy (units/day), and length of hospital stay, all of which were considered to be related to swallowing function or nutritional status [2,8,14,47,48,49]. Baseline values (at admission) for each outcome were also included as adjustment factors for each multivariate analysis. To reduce bias, common confounding factors were employed and adjusted for in a series of multivariate analyses. Finally, a variance inflation factor value of less than 10 was considered, as there was no multicollinearity in each analysis.

### 2.9. Ethics

This study was conducted in compliance with the Declaration of Helsinki and the Ethical Guidelines for Medical and Health Research Involving Human Subjects. The Institutional Review Board of the hospital where the study was conducted approved this study in advance (approval number: 199-220909). An opt-out procedure allowed participants to withdraw from the study at any time. Written informed consent could not be obtained due to limitations, resulting from the retrospective study design.

## 3. Results

A total of 1018 stroke patients were hospitalized during the study period. Among them, patients with missing data (n = 280), altered consciousness (n = 36), pacemakers (n = 36), or who were transferred to other hospitals (n = 80) were excluded. Then, a total of 586 patients were screened, of whom 257 were diagnosed with sarcopenia on admission and, therefore, included in the final analysis (Figure 1).

Summary of baseline characteristics of subjects enrolled and analyzed in the study are shown in Table 1. The included patients had a mean age of 79.3 (10.0) years, and 44.0% of the participants were male. At admission, the median FILS score was 7 (2–9), and 104 patients (40.4%) had dysphagia with an FILS score of 6 or less. The median number of medications prescribed at admission was 5 (3–7), and polypharmacy was present in 63.4% of patients. In between-group comparisons, baseline stroke history rate, premorbid mRS score, BMI, and number of PIMs were significantly higher, while FIM-cognitive scores were significantly lower in patients with polypharmacy than in those without polypharmacy.

Table 2 summarizes the details regarding the prescription at admission. The most frequently prescribed medications were antihypertensive drugs, antithrombotics, proton pump inhibitors (PPI), statins, antidiabetics, and diuretics.

Table 3 shows a comparison of the outcomes of the two groups according to the presence of polypharmacy at admission. Patients with polypharmacy at admission had significantly lower FILS scores (9 (7–10) vs. 10 (9–10), *p* = 0.004) at discharge than patients without polypharmacy at admission. There was no significant difference in the secondary outcome GNRI between the two groups (94.4 (87.5–98.3) vs. 91.7 (87.6–98.5), *p* = 0.616).

Table 4 shows a multivariate linear regression analysis of study outcomes adjusted for potential confounders. No multicollinearity was found among the variables. Results showed that the total number of medications at admission was independently associated with FILS (β = −0.133, *p* = 0.017) and GNRI (β = −0.145, *p* = 0.003) at discharge, respectively.

## 4. Discussion

In the present study, we examined the impact of polypharmacy on dysphagia and nutritional status in post-stroke patients with sarcopenia. We found two new findings among post-stroke patients with sarcopenia requiring rehabilitation. (1) A high number of medications prescribed at admission was associated with dysphagia at discharge. (2) A high number of medications prescribed at admission was associated with malnutrition at discharge.

A high number of medications prescribed at admission was associated with dysphagia at discharge. To the best of our knowledge, this could be the first study to examine the association between polypharmacy and the swallowing function in patients with sarcopenia. We suggest that polypharmacy may contribute to dysphagia in post-stroke patients with sarcopenia, in addition to the effects of the stroke itself, the loss of muscle mass, and muscle strength due to sarcopenia. Polypharmacy is associated with xerostomia [23] and cognitive decline, [50] and may also affect swallowing function. In polypharmacy, anticholinergic load is associated with dysphagia [26,27], cognitive decline [51], and poor oral health [52]. Polypharmacy is a potentially high-risk factor for a high anticholinergic load [53]. The impact of polypharmacy should be considered when providing rehabilitation for dysphagia and individualized prescription optimization should be provided for correctable polypharmacy. For this purpose, a multidisciplinary team led by a pharmacist is warranted.

A high number of medications prescribed at admission was associated with malnutrition at discharge. Drugs are one of the factors contributing to malnutrition and drug-induced nausea and vomiting, diarrhea, xerostomia, and taste disturbances are factors that contribute to decreased nutritional intake [54]. Previous studies have shown that hyperpolypharmacy with more than 10 drugs is associated with undernutrition at 3 years in the older category [55]. Similarly, in hospitalized patients with sarcopenia, polypharmacy has been suggested to be associated with nutritional status. The core of treatment for sarcopenia focuses on exercise and nutritional therapy, and polypharmacy may be an inhibitor of the latter. It has also been reported that deprescribing is associated with increased nutritional intake in sarcopenic patients with polypharmacy [56]. To provide adequate nutritional therapy to patients with sarcopenia, the impact of polypharmacy should be considered, and prescriptions for medications that affect nutritional intake should be reviewed.

On the same note, it is important to review polypharmacy and optimize drug prescribing for patients with sarcopenia, regardless of the presence of stroke. Sarcopenia may increase the risk of adverse drug events due to decreased ADL and changes in body composition, such as decreased muscle mass. It is known to be associated with higher toxicity of molecularly targeted drugs in cancer patients [57]. In the current study, the number of drugs being taken at admission was associated with dysphagia and malnutrition at discharge in hospitalized patients with sarcopenia. Stroke patients can easily become polypharmacy patients if they have underlying conditions, such as hypertension, dyslipidemia, and diabetes mellitus, as well as complications, such as disinhibition, insomnia, dysuria, spasticity, and pain, each of which can result in the introduction of pharmacotherapy. Among the drugs listed in Table 2, antipsychotics [24] and benzodiazepines [58] may have negative effects on the swallowing function. Antithrombotic drugs, antihypertensive drugs, statins, and antidiabetic drugs are frequently prescribed for secondary stroke prevention purposes. In addition, PPIs, diuretics, antipsychotics, benzodiazepines, non-steroidal anti-inflammatory drugs, and antidepressants are also commonly prescribed for medical management after stroke. However, while they may affect the swallowing function, as well as nutritional intake, all of these drugs can be (and should be) considered for deprescribing after acute care. Considering the possibility that patients with polypharmacy on admission may have difficulty maximizing improvement not only in physical and cognitive function but also in swallowing and nutritional status, it is necessary to review prescription drugs, taper or discontinue them, and consider non-pharmacological treatment when normal rehabilitation is ineffective or when drug-induced functional decline is suspected [59]. To this end, a multidisciplinary team led by a pharmacist is needed to manage drug-related risks.

Despite the novelty of our findings, the study had some limitations. At first, since this was a retrospective cohort study conducted at a single center in Japan, there are limitations in the generalizability of the present findings. Second, it does not take into account the effects of individual medications. Antipsychotics, benzodiazepines, and anticholinergic agents affect the swallowing function and nutritional status. Conversely, angiotensin converting enzyme inhibitors [60], amantadine [61], cilostazol [62], and Banxia Houpo Tang [63] induce a swallowing reflex and prevent the onset of pneumonia. Mirtazapine [64] and Rokunjitou [65] also increase appetite. The evaluation of the number of drugs alone fails to take into account the impact of drugs that enhance these functions. It is also unclear whether outcomes improve when polypharmacy is eliminated. Further high-quality research that considers the effects of each medication in more detail is desirable.

In conclusion, polypharmacy was associated with dysphagia and malnutrition in post-stroke patients with sarcopenia. In addition to the combination of nutritional and exercise therapies, a review of prescription medications is important for the treatment of sarcopenia.

## Figures and Tables

**Figure 1 nutrients-14-04251-f001:**
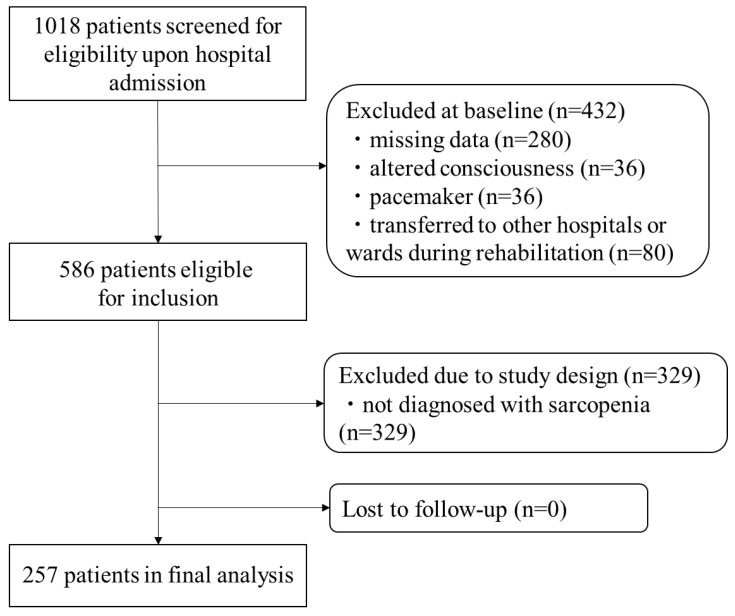
A flowchart of participant screening, inclusion, exclusion, and follow-up.

**Table 1 nutrients-14-04251-t001:** Summary of baseline subject information and comparison between the two groups by presence or absence of polypharmacy at admission.

	Total (N = 257)	Polypharmacy (+) (N = 163)	Polypharmacy (−) (N = 94)	*p* Value
Age, y	79.3 (10.0)	79.7 (9.0)	78.5 (11.5)	0.346
Sex, male	113 (44.0)	77 (47.2)	36 (38.3)	0.192
Stroke type				
Cerebral infarction	166 (64.6)	110 (67.5)	56 (59.6)	0.224
Cerebral hemorrhage	78 (30.4)	44 (27.0)	34 (36.2)	0.159
Subarachnoid hemorrhage	12 (4.7)	8 (4.9)	4 (4.3)	0.999
Stroke history	80 (31.1)	61 (37.4)	19 (20.2)	0.005
Premorbid mRS	1 (0, 3)	1 (0, 3)	0 (0, 2)	0.001
Onset-admission days	14 (10, 22)	15 (10, 25)	13 (9, 19)	0.074
Paralysis				
Right/Left/Both	114 (44.4)/99 (38.5)/17 (6.6)	75 (46.0)/57 (35.0)/13 (8.0)	39 (41.5)/42 (44.7)/4 (4.3)	0.516/0.144/0.305
BRS				
Upper limb/Hand-finger/Lower limb	4 (2, 6)/4 (2, 5)/5 (2, 6)	4 (2, 6)/4 (2, 6)/5 (2, 6)	4 (2, 5)/4.5 (2, 5)/4 (2, 5)	0.895/0.725/0.757
FIM, score				
-Total	43 (24, 73)	40 (23, 69)	46 (29, 80)	0.105
-Motor	25 (14, 53)	23 (13, 52)	27 (14, 55)	0.249
-Cognitive	15 (9, 24)	14 (8, 23)	19 (10, 25)	0.014
FILS, score	7 (2, 9)	7 (2, 9)	7 (7, 10)	0.121
ROAG, score	11 (10, 14)	12 (10, 15)	11 (9, 13)	0.117
CCI, score	3 (1, 4)	3 (2, 4)	3 (1, 3)	0.189
Nutritional status				
GNRI	89.5 (83.2, 96.4)	90.4 (84.0, 96.8)	88.0 (81.3, 95.4)	0.103
BMI, kg/m^2^	20.5 (18.4, 22.6)	21.2 (19.1, 22.9)	19.8 (13.7, 31.0)	0.013
Energy intake, kcal/kg/day	28.8 (24.5, 34.1)	28.3 (24.2, 33.7)	29.7 (25.1, 35.1)	0.259
Protein intake, g/kg/day	1.11 (0.95, 1.30)	1.09 (0.95, 1.23)	1.12 (0.94, 1.35)	0.128
Muscle-related variables				
HG, kg	12.6 (5.4, 17.8)	12.7 (5.3, 17.7)	12.5 (6.0, 17.9)	0.964
SMI, kg/m^2^	5.17 (4.63, 5.95)	5.22 (4.73, 6.06)	5.10 (4.37, 5.78)	0.067
Laboratory data				
Alb, g/dL	3.41 (0.50)	3.41 (0.54)	3.42 (0.43)	0.895
CRP, mg/dL	1.40 (2.52)	1.59 (2.84)	1.07 (1.79)	0.113
Hb, g/dL	12.71 (1.69)	12.58 (1.78)	12.95 (1.50)	0.092
Length of stay, days	104 (71, 145)	101 (67, 144)	107 (72, 147)	0.465
Rehabilitation ^a^, units/day	8.1 (7.0, 8.5)	8.0 (6.9, 8.4)	8.3 (7.2, 8.6)	0.498
Medication				
No. of total medications	5 (3, 7)	7 (5, 9)	3 (2, 3)	<0.001
No. of PIMs	1 (0, 1)	1 (1, 2)	0 (0, 1)	<0.001

Alb, albumin; BMI, body mass index; BRS, Brunnstrom recovery stage; CCI, Charlson’s comorbidity index; CRP, C-reactive protein; FILS, Food Intake Level Scale; FIM, Functional Independence Measure; GNRI, geriatric nutritional risk index; Hb, hemoglobin; HG, handgrip strength; mRS, Modified Rankin Scale; ROAG, Revised Oral Assessment Guide; PIMs, potentially inappropriate medications; SMI, skeletal muscle mass index. ^a^ Rehabilitation therapy performed during hospitalization (1 unit = 20 min).

**Table 2 nutrients-14-04251-t002:** Categories of medications prescribed on admission.

Drug Category	
Antithrombotics	165 (64.2)
PPI	145 (56.4)
Antihypertensives	168 (75.4)
Diuretic	42 (16.3)
Statins	71 (27.6)
Antidiabetics	42 (16.3)
Antipsychotics	18 (7.0)
Benzodiazepines	16 (6.2)
NSAIDs	16 (6.2)
Antidepressants	15 (5.8)

NSAIDs, non-steroidal anti-inflammatory drugs; PPI, proton pump inhibitors.

**Table 3 nutrients-14-04251-t003:** Bivariate analysis of outcomes between two groups with and without polypharmacy.

	Total (N = 257)	Polypharmacy (+) (N = 163)	Polypharmacy (−) (N = 94)	*p* Value
FILS at discharge, score	9 (8, 10)	9 (7, 10)	10 (9, 10)	0.004
GNRI at discharge	93.4 (87.6, 98.6)	94.4 (87.5, 98.3)	91.7 (87.6, 98.5)	0.616

FILS, Food Intake Level Scale; GNRI, geriatric nutritional risk index.

**Table 4 nutrients-14-04251-t004:** Multivariate linear regression analysis of study outcomes (FILS and GNRI) at discharge in post-stroke hospitalized patients with sarcopenia.

	FILS at Discharge			GNRI at Discharge		
	β	B (95% CI)	*p* Value	β	B (95% CI)	*p* Value
Age	−0.173	−0.042 (−0.071, −0.014)	0.004	−0.188	−0.171 (−0.264, −0.077)	<0.001
Sex (Male)	−0.106	−0.532 (−1.336, 0.271)	0.193	−0.145	−2.737 (−5.421, −0.052)	0.046
Stroke type						
Cerebral infarction	0.172	0.888 (−0.340, 2.117)	0.156	−0.112	−2.150 (−6.300, 2.000)	0.308
Cerebral hemorrhage	0.182	0.972 (−0.300, 2.243)	0.134	−0.152	−3.006 (−7.295, 1.283)	0.168
Subarachnoid hemorrhage	(reference)			(reference)		
FIM-motor on admission	0.043	0.005 (−0.017, 0.027)	0.642	−0.080	−0.038 (−0.111, 0.036)	0.313
FIM-cognitive on admission	0.028	0.008 (−0.039, 0.055)	0.730	0.024	0.026 (−0.130, 0.183)	0.741
FILS on admission	0.325	0.264 (0.132, 0.396)	<0.001	−0.050	−0.150 (−0.584, 0.285)	0.497
GNRI on admission	0.137	0.034 (0.002, 0.066)	0.038	0.660	0.631 (0.526, 0.737)	<0.001
Energy intake on admission	0.040	0.012 (−0.024, 0.047)	0.518	−0.004	−0.004 (−0.118, 0.110)	0.944
ROAG on admission	−0.008	−0.005 (−0.102, 0.091)	0.912	−0.043	−0.125 (−0.475, 0.224)	0.480
HG on admission	0.254	0.077 (0.028, 0.125)	0.002	0.065	0.073 (−0.082, 0.227)	0.353
SMI on admission	0.005	0.013 (−0.414, 0.440)	0.953	0.187	1.757 (0.374, 3.141)	0.013
CCI	−0.078	−0.112 (−0.271, 0.047)	0.165	0.084	0.468 (−0.071, 1.007)	0.088
Length of stay	0.092	0.005 (−0.002, 0.012)	0.163	0.083	0.017 (−0.006, 0.040)	0.139
Rehabilitation ^a^	0.025	0.018 (−0.063, 0.100)	0.658	−0.011	−0.029 (−0.284, 0.225)	0.821
Number of Drugs on admission	−0.133	−0.115 (−0.210, −0.021)	0.017	−0.145	−0.471 (−0.783, −0.159)	0.003
R^2^		0.480			0.608	

CCI, Charlson’s comorbidity index; FILS, Food Intake Level Scale; FIM, Functional Independence Measure; GNRI, geriatric nutritional risk index, HG, handgrip strength; ROAG, Revised Oral Assessment Guide; PIMs, potentially inappropriate medications; SMI, skeletal muscle mass index. ^a^ Rehabilitation therapy performed during hospitalization (1 unit = 20 min).

## Data Availability

The data are not publicly available owing to opt out restrictions. Data sharing is not applicable.

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
