# Peer review of "Polypharmacy and Its Association with Dysphagia and Malnutrition among Stroke Patients with Sarcopenia"

_nutrients, 2022, doi:10.3390/nu14204251_

Round 1

Reviewer 1 Report

This study assessed the association between number of medications and severity of dysphagia and malnutrition in stroke patients with sarcopenia undergoing rehabilitation, showing that polypharmacy is correlated with decreased oral intake. These results could help inform medication management in these vulnerable and often elderly patients, in whom side effects of polypharmacy have been well documented. The focus on dysphagia adds a novel aspect to this ongoing area of investigation. However, as a retrospective analysis, future prospective studies to evaluate the effectiveness of possible interventions would be necessary for any definitive recommendations regarding clinical practice. The presentation of results, statistical analysis, and use of English language are excellent.

Minor comment:

1. Discussion, ln 243-244: According to the results, polypharmacy was associated with significantly lower FILS. Therefore, because lower FILS represents greater severity of dysphagia, it would be appropriate to say “The number of medications at admission was negatively associated with oral intake at discharge” or “The number of medications at admission was positively associated with dysphagia at discharge”, but the current phrasing implies the opposite of this. Ln 248 and 299 should similarly be rephrased.

Grammar:

1. Abstract, ln 28: “Pre-scription” should be changed to “prescription”.

2. [2.4. Drug information], ln 130: “PIMs” should be within parentheses. 

Author Response

Reviewer 1

This study assessed the association between number of medications and severity of dysphagia and malnutrition in stroke patients with sarcopenia undergoing rehabilitation, showing that polypharmacy is correlated with decreased oral intake. These results could help inform medication management in these vulnerable and often elderly patients, in whom side effects of polypharmacy have been well documented. The focus on dysphagia adds a novel aspect to this ongoing area of investigation. However, as a retrospective analysis, future prospective studies to evaluate the effectiveness of possible interventions would be necessary for any definitive recommendations regarding clinical practice. The presentation of results, statistical analysis, and use of English language are excellent.

(Response)

We sincerely appreciate your supportive and positive comments, which really helped us improve our manuscript.

Minor comment:

  1. Discussion, ln 243-244: According to the results, polypharmacy was associated with significantly lower FILS. Therefore, because lower FILS represents greater severity of dysphagia, it would be appropriate to say “The number of medications at admission was negatively associated with oral intake at discharge” or “The number of medications at admission was positively associated with dysphagia at discharge”, but the current phrasing implies the opposite of this. Ln 248 and 299 should similarly be rephrased.

(Response)

Thanks for the comment. We agree. We have revised it by removing "negatively".

Grammar:

  1. Abstract, ln 28: “Pre-scription” should be changed to “prescription”.

(Response)

Thanks for the comment. We have revised “Pre-scription” to “prescription”.

  1. [2.4. Drug information], ln 130: “PIMs” should be within parentheses.

(Response)

Thanks for the comment. We have revised it by putting "PIMs" in parentheses.

Reviewer 2

The problem(s) of polypharmacy in post-stroke patients is convincingly described, although well known. However, as the authors discuss under "Limitations of the Study" the effects of individual medications remain completely unanswered! What is the consequence: Shall these patients remain completely untreated (would be of ethical concern) - resp.  what medications should (must?) be (dis)continued? Antidiabetics? Antihypertensives? Antibiotics? Authors must at least try to make a sound proposal. Otherwise, the reader is still confronted with an unresolved problem!

(Response)

Thanks for the comment. We agree. While there are some drugs that can negatively affect swallowing function and nutritional status, it is also clear that there is an impact of polypharmacy. However, it is unclear from this study which drugs should be continued or considered for drug discontinuation. Therefore, we have created a new Table 2, examining the frequency of use in our enrolled patients of drugs that may be commonly prescribed to post-stroke patients. We have added a discussion of drugs that need to be continued and those that affect swallowing function, arousal, and nutritional intake and can be considered for discontinuation.

(Change)

(New 3rd paragraph in the Result and New Table 2)

Table 2 summarizes the details regarding the prescription at admission. The most frequently prescribed medications were antihypertensive drugs, antithrombotics, proton pump inhibitors (PPI), statins, antidiabetics, and diuretics.

Table 2. Categories of medications prescribed on admission

Drug category

Antithrombotics

165 (64.2)

PPI

145 (56.4)

Antihypertensives

168 (75.4)

Diuretic

42 (16.3)

Statins

71 (27.6)

Antidiabetics

42 (16.3)

Antipsychotics

18 (7.0)

Benzodiazepines

16 (6.2)

NSAIDs

16 (6.2)

Antidepressants

15 (5.8)

NSAIDs, non-steroidal anti-inflammatory drugs; PPI, proton pump inhibitors.

(4th paragraph in the Discussion)

“On the same note, reviewing polypharmacy is important for patients with sarco-penia undergoing rehabilitation. Sarcopenia may increase the risk of adverse drug events due to decreased ADL and changes in body composition, such as decreased muscle mass. It is known to be associated with higher toxicity of molecularly targeted drugs in cancer patients [59]. In this study, the number of drugs being taken at admis-sion was associated with dysphagia and nutritional risk at discharge in hospitalized patients with sarcopenia after stroke. Stroke patients can easily become polypharmacy patients if they have underlying conditions such as hypertension, dyslipidemia, and diabetes mellitus, as well as complications such as disinhibition, insomnia, dysuria, spasticity, and pain, each of which can result in the introduction of pharmacotherapy. Among the drugs listed in Table 2, antipsychotics [24] and benzodiazepines[60] may have negative effects on swallowing function. Antithrombotic drugs, antihypertensive drugs, statins, and antidiabetic drugs are frequently prescribed for secondary stroke prevention purposes. In addition, PPIs, diuretics, antipsychotics, benzodiazepines, non-steroidal anti-inflammatory drugs, and antidepressants are also commonly pre-scribed for medical management after stroke. However, while they may affect swal-lowing function as well as nutritional intake, all of these drugs can be (and should be) considered for deprescribing after acute care. Considering the possibility that patients with polypharmacy on admission may have difficulty maximizing improvement not only in physical and cognitive function but also in swallowing and nutritional status, it is necessary to review prescription drugs, taper or discontinue them, and consider non-pharmacological treatment when normal rehabilitation is ineffective or when drug-induced functional decline is suspected [61]. To this end, a multidisciplinary team led by a pharmacist is needed to manage drug-related risks.”

Reviewer 2 Report

The problem(s) of polypharmacy in post-stroke patients is convincingly described, although well known . However, as the authors discuss under "Limitations of the Study" the effects of individual medications remains completely unanswered  ! What is the consequence: Shall these patients remain completely  untreated (would be of ethical concern) - resp.  what medications should (must?) be (dis)continued? Antidiabetics? Antihypertensives? Antibiotics? Authors must at least try to make a sound proposal. Otherwise the reader is still confronted with an unresolved problem!

Author Response

(The authors gave the same response as above.)

Round 2

Reviewer 2 Report

The additionally provided informations are valuable, have significantly improved the paper and now provide important data for the practioner